# Exploring How Stereotype Modification Mediates the Relationship between Social Dominance and Multicultural Acceptance

**DOI:** 10.3390/bs14090745

**Published:** 2024-08-26

**Authors:** Sowon Lee, Boyoung Kim

**Affiliations:** College of Nursing, Chonnam National University, Gwangju 61469, Republic of Korea; sowonlecture@gmail.com

**Keywords:** social dominance orientation (SDO), discrimination, multicultural acceptance, cognitive flexibility

## Abstract

This research examines the mediating role of stereotype modification in the relationship between social dominance orientation (SDO) and multicultural acceptance in South Korea’s increasingly diverse society. We obtained a sample of 402 participants between the ages of 20 and 65 through an online survey. We used SPSS 26.0 for statistical analyses, including frequency, correlation, and regression analyses. Employing Hayes’ Model 4, we examined the mediation effect with a bootstrap sample of 10,000 iterations, determining the significance of the effect with a 95% confidence interval (CI). The results revealed nuanced relationships among the variables, shedding light on the complex dynamics of social cognition and intergroup relationships in the South Korean context. The research concludes that individuals with a higher social dominance orientation tend to have lower acceptance of multiculturalism and are more likely to hold prejudiced attitudes toward outgroups. This finding suggests that SDO is a significant factor in integrating and adapting migrants into host societies and can lead to social conflict. The study implies that addressing SDO is crucial for fostering positive attitudes toward multiculturalism and reducing discrimination.

## 1. Introduction

In the age of globalization, people no longer belong to just one culture but are influenced by various cultures [1]. The encounter between diverse cultures has resulted in significant advancements, including technological progress and the integration of the global economy [2], but it has also generated conflict, hatred, and fear due to perceptions of foreign cultures, leading to various forms of discrimination based on race, gender, religion, and other factors [3]. To address these issues, the United Nations adopted the Genocide Convention and the Convention on the Elimination of All Forms of Racial Discrimination, which prohibit discrimination based on race [4]. In addition, the United States enacted the Civil Rights Act, and Australia enacted the Racial Hate Act, prohibiting discrimination based on race, color, religion, sex, and ethnic origin [5]. In addition, the United Kingdom fined a spectator who mimicked gouging out an athlete’s eyes to humiliate a South-Korean-born athlete and banned them from attending a game [6], and other countries around the world are making various efforts to prevent discrimination. These international examples underscore the importance of legal and social measures to promote equality and provide insights for improving policies and social attitudes in South Korea.

South Korea is also rapidly becoming a multicultural society due to the influx of foreign workers and international marriage migration, emphasizing the significance of understanding multiculturalism [7]. However, South Korea is constantly discussing bills to prohibit discrimination against minority groups but has not yet enacted any legislation [8]. This lack of legislation is because South Koreans are less likely to come into direct contact with foreigners or immigrants daily than residents of other countries with anti-discrimination laws, leading to a perceived lack of necessity for such laws [9]. Although South Korea uses a positive image of multiculturalism, it treats foreigners as objects of assimilation or exclusion [3]. This attitude led to the 2022 South Korean Consciousness and Values Survey, where 67.4% of respondents responded that there is still prejudice against foreigners based on their race and country [10]. Discrimination against foreign migrants, lack of preparation, and experience in multicultural societies can hinder migrants’ adaptation and integration into the host social environment, leading to social conflict between migrants and the host population [11]. Because of prejudices against migrants, the migrants perceive South Korea as a society with high levels of discrimination and barriers to integration [12].

Discrimination and prejudice are individual expressions of intergroup conflicts [13,14]. From the perspective of intergroup conflict, mainstream society perceives migrants as an “outgroup” and, therefore, subject to conflict. Those high in social dominance orientation (SDO) are more likely to perceive migrants as outgroups [15]. SDO strongly predicts intergroup attitudes and behaviors and social and political attitudes. Pratto and colleagues [16] defined SOD as “a general attitudinal orientation toward intergroup relations that reflects the degree to which one prefers to view intergroup relations as egalitarian or hierarchical”. In other words, SDO is the tendency to believe that one’s group is superior to others with a desire to dominate others. SDO is an individual-level psychological component of social dominance theory [17], which posits that our social world organizes around a group-based hierarchy of dominant and subordinate groups.

Social groups use various social variables to organize, such as culture and nationality. SDOs are susceptible to socialization and prolonged exposure to certain social environments [18]. For example, individuals exhibit higher levels of SDO when socialized in contexts characterized by high levels of inequality and competition for power and status [19]. South Korean studies that examined the relationship between SDO and multicultural acceptance [15,20] found an association between a stronger SDO and lower multicultural acceptance. Research on social dominance orientation and xenophobia has shown that people with stronger social dominance orientation have more aggressive and hostile attitudes toward illegal immigrants [21]. Based on these findings, we can predict that people with high social dominance orientation tend to have conservative traits and are more likely to exclude or ostracize rather than embrace multiculturalism and other races. In contrast, a more progressive orientation correlates with more positive attitudes toward change in a multicultural society and acceptance of multicultural members.

As a relatively stable individual difference, SDO strongly predicts stereotyping, discrimination, and prejudice [17,18,22,23]. SDO is associated with a greater tendency to endorse and maintain social hierarchies and inequalities, which involves less flexibility in altering one’s stereotypes [24]. High-SDO individuals are typically resistant to changing their ingrained stereotypes and are more likely to exhibit prejudice toward low-status groups [25]. Conversely, there is a link between lower levels of SDO and greater cognitive flexibility, which includes a higher propensity for empathy and cooperation with diverse groups [15,26]. Research indicates an inverse relationship between cognitive flexibility, the ability to adapt one’s thinking and behavior to new or changing situations [27,28], and SDO’s rigidity. Specifically, individuals with high SDO often show lower cognitive flexibility, contributing to their resistance to altering negative stereotypes [25].

In contrast, individuals with lower SDO levels, who are generally more open and less rigid in their thinking, exhibit greater cognitive flexibility, which enhances their ability to adopt more positive attitudes toward minority groups and adjust to diverse social contexts [29,30]. This relationship suggests that increasing cognitive flexibility may mitigate some of the negative impacts of high SDO, potentially leading to more accepting attitudes toward outgroups and reduced prejudice [29,31]. Therefore, fostering cognitive flexibility could effectively counter the biases associated with high SDO and promote more inclusive and adaptable social attitudes.

This attitude of helping or supporting others to coexist as equals within their culture without prejudice against the culture to which they belong is called multicultural acceptance [32]. Multicultural acceptance is an attitude that recognizes the transition to a multicultural society as positive and supports the social value of coexisting with different ethnic and racial groups [33,34]. In South Korea, the Ministry of Gender Equality and Family conducts a multicultural acceptance survey every three years to raise public awareness of multiculturalism and determine multicultural education’s direction. According to the findings of the 2021 National Multicultural Acceptance Survey [35], the level of multicultural acceptance varies by age. The survey revealed that the younger the adolescents and adults, the higher their level of multicultural acceptance. In domestic studies, multicultural receptivity is “multicultural acceptance”, but there is limited research on this concept in international contexts.

Most studies predominantly examine multicultural politics from a white perspective [36], focusing on concepts like “multicultural attitudes” or “attitudes toward multiculturalism” [37,38]. In the case of South Korea, despite the increasing rate of foreign immigration, the country remains predominantly mono-ethnic, leading to prejudices and discrimination against certain nationalities or races [39]. The many roles that stereotype-changing tendencies and cognitive flexibility play are critical to understand in order to reduce conflicts throughout the shift to a multicultural society. External factors including social conventions, education, the media, and one’s group membership might have an impact on a person’s tendency to change stereotypes [40]. These tendencies refer to the propensity of individuals to change their stereotypes in response to changing social expectations or new information. On the other hand, cognitive flexibility refers to an individual’s inherent abilities and dispositions, including brain function, cognitive skills, and personal experiences [41]. Cognitive flexibility is the ability to adapt one’s thinking and behavior in response to new or unexpected conditions, independent of external influences [42]. Understanding the independent relationship between these two constructs is essential for assessing how interactions with SDO might affect intercultural acceptance. While tendencies to change stereotypes can shift based on external inputs and social pressures, cognitive flexibility is a more intrinsic trait that influences how individuals process and integrate new cultural information [43]. Thus, this study focuses on the tendency to change stereotypes and cognitive flexibility as mediators of multicultural acceptance. We expect the results to affect future multicultural policy and multicultural education. Figure 1 illustrates this study’s framework.

## 2. Method

### 2.1. Study Design

This descriptive correlational study examines the relationship between social dominance orientation and multicultural acceptance, focusing on how stereotype modification and cognitive flexibility mediate this relationship.

### 2.2. Participants

We conducted this study among South Korean adults aged 20 to 65 who understood the study’s purpose and methodology and agreed to participate. We used G-Power 3.1 to calculate the required number of subjects. Based on a significance level of 0.05, a power of 95%, ten explanatory variables, and a medium effect size of 0.15, the minimum sample size required was 178 participants. Given the online survey dropout rate and a 30% attrition rate to account for random responses, we selected 402 participants for the study. The characteristics of the participants in this study are in Table 1.

### 2.3. Measures

#### 2.3.1. Social Dominance

We used the South Korean version of the SDO scale, originally from Pratto et al. [16] but adapted by Lee and Yoo [44]. The SDO scale comprises two subfactors: dominance, which favors dominating others, and anti-egalitarianism, which favors inequality in intergroup relations. The SDO scale comprises 16 items, eight items for each subfactor. The scale uses a seven-point Likert-type measure, with higher scores indicating greater social dominance orientation. Example items are ‘Some groups of people are simply inferior to other groups’ and ‘All groups should receive an equal chance in life’. The reliability of the original instrument was 0.76 for the dominance domain and 0.87 for the inequality sub-domain. In this study, the reliability was 0.87.

#### 2.3.2. Multicultural Acceptance

We measured multicultural acceptance using the Multicultural Attitude Scale Questionnaire (MASQUE) [45], adapted and validated by Kang and Lim [46]. It is a 16-item, six-point scale consisting of three factors: six items on difference perception, five on openness and acceptance, and five on commitment to action. Sample items are ‘I am emotionally concerned about racial inequality’ and ‘People’s social status does not affect how I care about them’. In Kang and Lim’s [46] study, the reliability was 0.79 for difference perception, 0.77 for openness and acceptance, and 0.78 for commitment to action, while this study’s reliability was 0.87.

#### 2.3.3. Stereotype Change Tendency

We used MacNab et al.’s [47] scale to measure the tendency to change stereotypes. The scale measures five items on a five-point Likert scale ranging from 1 (strongly disagree) to 5 (strongly agree). Example items include ‘I am open to changing my stereotypes’ and ‘I think the ability to modify one’s beliefs about other groups is important’. Cronbach’s alpha of the original instrument was 0.84, and the reliability in this study was 0.89.

#### 2.3.4. Cognitive Flexibility

We measured cognitive flexibility using the Cognitive Flexibility Inventory (CFI) developed by Dennis and Vander Wal [48] and adapted and validated by Heo, Shim, and Yang [49]. It is a 19-item instrument consisting of two subscales: alternative and control. The measure uses a seven-point Likert-type scale, with higher scores indicating greater cognitive flexibility. Sample items include ‘I consider multiple options before making a decision’ and ‘It is important to look at difficult situations from many angles’. Cronbach’s ɑ of the original instrument was 0.86, and in Choi et al.’s [30] study, it was 0.81. The reliability in the present study was 0.80.

### 2.4. Data Collection and Research Ethics

The research ethics committee of Chonnam National University approved this study. We collected the data for three days, from 8 to 14 February 2023, using an online survey company. We only targeted adults who read the information about the purpose of the study and agreed to participate. We added trap questions in the middle of the survey to prevent random answers, and we terminated the survey if we did not receive the appropriate answer (e.g., if we received random answers). We rewarded points to participants who completed the survey that they could redeem on the investigator’s research site. The study comprised 402 participants, and the survey took approximately 20 min.

### 2.5. Data Analysis

This study used SPSS 26.0 for frequency, correlation, and regression analyses. In addition, we used Hayes’ [50] Model 4 to test the mediation effect. We tested the significance of the mediation effect with a bootstrap sample of 10,000 to analyze the 95% confidence interval (CI).

## 3. Results

### 3.1. Bivariate Correlation Analysis

The results of the correlation analysis between the variables used in the study are in Table 2. Social dominance orientation was negatively related to multicultural acceptance and stereotype change beliefs. On the other hand, multicultural acceptance, stereotype change beliefs, and cognitive flexibility were positively related.

### 3.2. Mediation Effect Analysis

Table 3 presents the results of the mediating effect of the tendency to change stereotypes and cognitive flexibility on the relationship between SDO and multicultural acceptance. We found significant results for the tendency to change stereotypes as a mediator but not for cognitive flexibility as a mediator. The paths for the independent variables of social dominance orientation, tendency to change stereotypes, and cognitive flexibility were all significant.

The total, direct, and indirect effects of the mediating effects of the tendency to change stereotypes and cognitive flexibility on the relationship between SDO and multicultural acceptance are in Table 4. The direct effect of multicultural acceptance on SDO is significant. The mediating effect of the tendency to change stereotypes is also significant, confirming that the tendency to change stereotypes partially mediates SDO and multicultural acceptance. The indirect effect of cognitive flexibility as a mediator was not significant. The analysis using controlled variables, specifically examining the impact of trip abroad experience and multicultural training experience on the mediating effects, resulted in a consistent conclusion that the tendency to change stereotype (TCS) remains a key mediator. The analysis results, including these controlled variables, are in Appendix A.

## 4. Discussion

This study examined the mediating effects of stereotype modification tendency and cognitive flexibility on the relationship between social dominance orientation (SDO) and multicultural acceptance. Results indicated that stereotype modification tendency partially mediated the relationship between SDO and multicultural acceptance, but cognitive flexibility did not.

SDO is an individual’s attitude toward discrimination and inequality in social groups, and individuals with a strong SDO may be more likely to hold group prejudicial ideologies or to hierarchize groups [16]. These tendencies may manifest as racism and nationalism and may favor the dominance of certain groups over other minority groups [51]. A review of the literature on dominance orientation reveals several studies that focus on social minority or gender effects, such as xenophobia [52], sexual minority and stigma associations [53], and political orientation and sexism [54]. The literature suggests that SDO affects attitudes toward other groups in the relationship between mainstream groups and minority groups, such as multicultural and lesbian, gay, bisexual, transgender, and queer (LGBTQ) groups.

Social identity theory [14,55] posits that individuals differentiate between “us” and “them” based on the relationship of the groups to issues of discrimination and exclusion when individuals seek to enhance their self-esteem by attributing positive qualities to their group. In this context, high social dominance orientation is discriminatory behavior toward other groups (multicultural, diverse minority groups) aimed at perceiving one’s ingroup as superior or better. This study found that individuals with higher SDO tended to have lower levels of multicultural acceptance. Given these findings, it is likely that strong discriminatory or exclusionary attitudes toward other groups are associated with negative attitudes toward multiculturalism and low acceptance.

In the past, multicultural policies emphasized the adaptation of migrants by categorizing them into multiculturalism, which recognizes different cultures, and assimilationism, integrating them into the mainstream culture [56]. Since the 2000s, most countries have adopted interculturalism as a model for the social integration of migrants. This approach emphasizes migrants’ adaptation, interaction, and engagement with the wider society’s culture, including the mainstream culture [57]. This interculturalism is possible when people do not have discriminatory attitudes such as stereotypes and prejudices [35], and reducing stereotypes and prejudices promotes an open and flexible attitude toward other cultures. Previous research has shown that individuals high in SDO are more likely to be prejudiced against lower-status groups. In contrast, individuals low in SDO are more likely to cooperate with other groups [15,26]. Studies have also reported that higher levels of openness and proactivity are associated with higher intercultural acceptance [58]. However, scholars have shown that stronger group identification with ethnicity or belonging negatively impacts intercultural acceptance [59,60].

Research indicates that SDO reflects prejudice against socially disadvantaged, minority, and low-status groups more than prejudice against competing groups [61]. In the case of South Korea, it may also influence people to hold negative attitudes toward minority groups, such as the belief that foreign workers take jobs from South Koreans [62]. These stereotypes and prejudices can hinder South Korea’s transformation into a multicultural society and its potential to become more inclusive. Countries like the United States, Canada, and Australia have laws prohibiting discrimination and prejudice and impose sanctions in cases of discrimination based on race, culture, and other factors. In South Korea, however, there is no legal basis for punishing discrimination based on race, culture, etc. Therefore, to increase multicultural acceptance, it is necessary to find ways to reduce prejudices and stereotypes rooted in social dominance orientation. It is important to promote awareness and education that challenge existing biases and emphasize the benefits of diversity. Since greater contact and experience with multicultural settings can help build a more inclusive and open society [63,64,65], it is essential to develop and enhance systems that expand multicultural experiences. However, research suggests that even with extensive multicultural experiences, negative encounters can increase prejudice [66]. Therefore, it is crucial to establish solutions that address both the quantity and quality of positive multicultural experiences.

Flexibility refers to coping with various situations [67], while cognitive flexibility pertains to applying acquired knowledge effectively in diverse contexts. Cognitive flexibility helps individuals adapt their thoughts and behaviors or consider alternative perspectives based on different situations [30]. In addition, cognitively flexible individuals demonstrate adaptability to changing environments and possess the adaptive ability to cope effectively in relationships with others [68]. Therefore, individuals with higher cognitive flexibility have fewer prejudices and are more receptive to different cultures. This study found a significant relationship between cognitive flexibility and multicultural acceptance. In addition, numerous studies have validated the relationship between cognitive flexibility and attitudes toward others [29,30].

However, the mediation of cognitive flexibility on the relationship between SDO and multicultural acceptance was not significant in this study. This finding contradicts our hypothesis that people high in SDO would show reduced cognitive flexibility because they have a closed mindset [69] and that cognitive flexibility would mediate and influence intercultural acceptance. We believe that these findings open several possibilities. First, SDO and cognitive flexibility may not always be inversely related. Palese and Schmid Mast [70] reported that individuals high in SDO have higher levels of behavioral flexibility, which is the ability to flexibly adapt one’s behavior to the individual situation and the needs of others. Martin and Heineber [71] also found that when leaders high in SDO are also high in empathy, transformational leadership—where leaders use flexible thinking to encourage colleagues to cooperate and change—is more likely than authoritarian leadership. These findings suggest that various factors influence SDO and cognitive flexibility, including an individual’s situation, environment, and personality. Therefore, it is necessary to clarify the relationship between SDO and cognitive flexibility by considering these factors.

Next, South Korea adheres to mononationalism, rooted in a strong sense of nationalism and collective identity [72]. Currently, South Korea is experiencing a shift toward a multicultural society, with the number of multicultural households increasing from 299,241 in 2015 to 415,584 in 2023 [73] and resident foreigners accounting for 4.89% of the total population [74]. From the perspective of previous research [75,76], which suggests that increased multicultural experiences affect SDO and cognitive flexibility, leading to greater multicultural acceptance, South Korea should theoretically exhibit lower SDO and higher cognitive flexibility, resulting in increased multicultural acceptance. However, these results may not hold in every case, considering that South Korea’s strong nationalist ideology might lead to high SDO and cognitive flexibility due to the fusion of nationalist ideology and multicultural experiences. This situation suggests that even multicultural contact experiences may not positively affect multicultural acceptance amid strong mononationalist beliefs. Therefore, although multicultural contact experiences are important, we must also consider South Korean nationalism. To increase multicultural acceptance, the government, citizens, and immigrants should collaborate to ensure that South Korean nationalism and multicultural society coexist and encourage positive multicultural experiences to raise awareness.

Because we conducted this study only among South Koreans, the results may reflect the cultural characteristics of South Korea. Therefore, it would be helpful to generalize the results of this study by conducting a follow-up study with different ethnic groups. In addition, it is necessary to redefine the causal relationship between variables through a thorough literature review to clarify the relationship between SDO and multicultural acceptance. We recommend conducting more in-depth studies to examine the mediating effects of the SDO subscales, SDO-D and SDO-E, on the relationship between SDO and multicultural acceptance. Finally, it is important to recognize that the tendency to change stereotypes and cognitive flexibility operate through different mechanisms. While cognitive flexibility is determined by an individual’s intrinsic abilities and tendencies, such as brain function, cognition, and experience, stereotypes are highly influenced by external variables such as social conventions, education, media, and an individual’s group [40,41]. According to Bedge, preconceptions can persist even in those with excellent cognitive flexibility [77]. This study indicates that stereotypes, rather than an individual’s internal cognitive abilities, have a greater influence on multicultural acceptance. Specifically, stereotype change was found to be a significant mediator of the relationship between SDO and multicultural acceptance, while cognitive flexibility was found to be an insignificant mediator. Therefore, multicultural policies, initiatives, etc. that accurately recognize the social and cultural trends and attitudes of the nation should be supplied in order to improve intercultural acceptance.

The present study is significant because it examines the mediating effects of stereotype-changing tendencies and cognitive flexibility on the relationship between SDO and multicultural acceptance. However, there are some limitations. Future research should extend to general populations of different ethnicities and countries. We also need richer methodologies that reflect the diversity of characteristics of individuals’ SDOs. In addition, we suggest that follow-up research include measures that reflect behavioral changes toward outgroups.

## 5. Conclusions

This study confirms that the tendency to change stereotypes mediates the relationship between social dominance orientation and multicultural acceptance. However, cognitive flexibility did not show such a mediating effect. This result implies that the higher the social dominance orientation, the lower the intercultural acceptance, consistent with multicultural policies in countries that emphasize the importance of interculturalism. Cognitive flexibility, which refers to adapting knowledge appropriately in different situations, was also related to intercultural competence. However, this flexibility did not mediate the relationship with social dominance orientation, suggesting that prejudice and stereotyping may not be solely cognitive issues. The results of this study contribute to understanding the relationship between SDO and multicultural acceptance in the South Korean social context. They also contribute to understanding multicultural societies and establishing policies for adapting social minorities and protecting their human rights.

## Figures and Tables

**Figure 1 behavsci-14-00745-f001:**
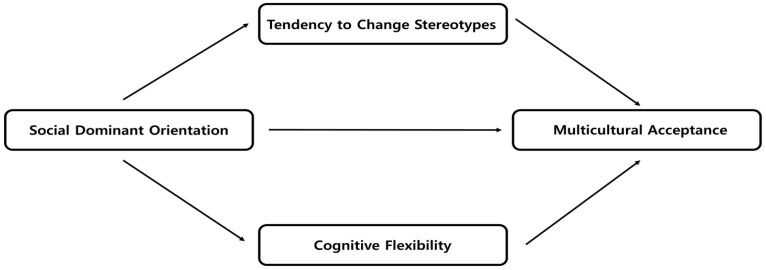
Study framework.

**Table 1 behavsci-14-00745-t001:** Participants’ general characteristics.

Variables	Categories	MN	(%)
Gender	Male	189	(47.0)
	Female	213	(53.0)
Age	19–29	82	(20.4)
	30–39	120	(29.8)
	40–49	118	(29.4)
	≥50	82	(20.4)
Number of trips abroad	No	79	(19.7)
	≥1	323	(80.3)
Multicultural training experience	No	281	(69.9)
	Yes	121	(30.1)
Education level	≤High school	85	(21.1)
	College	293	(72.9)
	≥Master	24	(6.0)
Religion	No	263	(65.4)
	Yes	139	(34.6)

**Table 2 behavsci-14-00745-t002:** Bivariate correlation analysis.

	M	SD	Social Dominance Orientation	Multicultural Acceptance r(*p*)	Stereotype Change Beliefs
Social dominance orientation	3.30	0.99			
Multicultural acceptance	3.63	0.63	−0.130 (0.009)		
Stereotype change beliefs	3.44	0.44	−0.136 (0.006)	0.445 (<0.001)	
Cognitive flexibility	3.33	0.35	0.067 (0.183)	0.391 (<0.002)	0.368 (0.001)

**Table 3 behavsci-14-00745-t003:** Mediation effect analysis.

		STEP1 (Model 1)				STEP2 (Model 2)				STEP3 (Model 3)			
		DV = TCS				DV = CFI				DV = MASQUE			
		β	SE	t	*p*	β	SE	t	*p*	β	SE	t	*p*
IV	SDO	−0.027	0.010	−2.755	0.0061	0.037	0.027	1.34	0.1826	−0.048	0.020	−2.360	0.0187
MV	TCS									0.777	0.110	7.030	<0.001
	CFI									0.235	0.040	5.946	<0.001
R^2^		0.137				0.067				0.518			
F (*p*)		7.590 (*p* < 0.01)				1.782 (*p* > 0.05)				48.533 (*p* < 0.001)			

Note. IV = independent variable; DV = dependent variable; MV = mediates variable; SDO = social dominance orientation; TCS = tendency to change stereotype; CFI = cognitive flexibility; MASQUE = multicultural acceptance.

**Table 4 behavsci-14-00745-t004:** Verifying the bootstrapping mediation effect.

SDO		Effect	SE	LLCI	ULCI	t	*p*
Total Effect		−0.060	0.023	−0.106	−0.015	−2.613	0.0093
Direct Effect		−0.048	0.020	−0.088	−0.008	−2.360	0.0187
Indirect Effect	SDO–TCS–MASQUE	−0.021	0.009	−0.042	−0.006		
	SDO–CFI–MASQUE	0.009	0.006	−0.003	0.022		

Note: SDO = social dominance orientation.

## Data Availability

Data are available upon substantiated request from the corresponding author.

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
