# Peer review of "Exploring How Stereotype Modification Mediates the Relationship between Social Dominance and Multicultural Acceptance"

_behavsci, 2024, doi:10.3390/bs14090745_

Round 1
Reviewer 1 Report
Comments and Suggestions for Authors
This study approaches the urgent and significant issue of multiculturalism respectively interculturalism from the perspective of the host society, in this case, South Korea, by means of the social dominance orientation (SDO) and its prevalence within the South Korean population. While study is ambitious in both its approach and literature review, there are some weaknesses which need to be rectified.
As a general rule, please write "South Korea" instead of "Korea". The reason is obvious. There are some references to the global situation which I do not estimate relevant for this context.
For further more concrete suggestions for improvement, please see attached file.

Author Response
Thank you for the opportunity to revise our manuscript. We appreciate your careful review and constructive suggestions. The suggested edits have substantially improved the manuscript. We’ve colored our modifications in blue. Please see the attachment.

Reviewer 2 Report
Comments and Suggestions for Authors
Comments for the author(s); behavsci-3093733
1. Thank you for the opportunity to review your manuscript. I carefully read it, with great interest. I think it has many positive attributes, was very interesting, and touches on a very important topic many societies (in Korea and beyond) are facing today. In particular, I found the idea that multicultural acceptance could be triggered by SDO via some form of stereotyping (but not cognitive flexibility) to be interesting. That said, I had some concerns that I think warrant addressing to fully realize the potential of this work.
THEORETICAL ISSUES:
2. I really enjoyed your introduction, but felt that it would help to more clearly differentiate this research from similar prior work.
In particular, I was very surprised to not see a few specific articles addressed in this work, given a lot of conceptual overlap with your research (and potentially conflicting findings or arguments).
For one, recent work suggests that negative multicultural experiences can alter SDO which, in turn, influences stereotyping (Affinito et al., 2023). Given your framing (e.g., tense intergroup relations stemming from migration, xenophobia, etc.), I was wondering if the OPPOSITE could be true – i.e., that negative interactions in a multicultural environment like Korea (which could trigger low levels of multicultural acceptance) can actually CHANGE people’s SDO. In other words, I think you need to more clearly argue for your model – or at least acknowledge this as a limitation given your results are correlational (i.e., meaning it is difficult to definitely say that the boxes and arrows point in the directions you specify).
To be clear, I am not suggesting that you need to conduct follow-up studies (although that would be ideal if possible). Just that other work potentially calls the causal order in your model into question.
Relatedly, a recent review of research on various multicultural experiences (such as the examples you draw on in your intro and discussion) suggest that positive experiences can actually ameliorate intergroup biases (e.g., Tadmor et al., 2012) because these sorts of interactions can be “cognitively liberating” (e.g., Hodson et al., 2018).
In other words, it seems like the exact type of experiences people have within multicultural environments (like Korea) may potentially change or drive some of the effects you hypothesize.
I’m not usually one to suggest incorporating other citations, but these seemed particularly relevant to at least acknowledge in your framing or mention as limitations of your data that future work could address.
3. Your model was interesting, but I think a bit more clarity of constructs would help. At times, I was confused about what precisely was being measured or hypothesized. For example, in your conclusion you write that “this study confirms that stereotype threat mediates…” (p. 8). I thought your mechanism was “tendency to change stereotypes” (Figure 1). Which one is it? I think being clear on this will help readers couch your paper in existing work. This may have just been a typo?
METHODOLOGICAL ISSUES:
4. Your study was well done, in my opinion, but I had some questions. For one, I’m confused about how you created your measures in Table 2, given the reported means. Did you sum the items? It could be personal preference, but I think it is more common to create composites based on averages of each item (vs. summing the items). Either way, I think it is important to disclose to readers how exactly you created scale composites from the items.
5. Did you control for any of the variables you report in Table 1? This seems important given your design is correlational, and prior work suggesting that different aspects of multicultural experiences (i.e., # of trips abroad; multicultural experience training) are related to variables in your model (e.g., Affinito et al., 2023 show multicultural experiences relate to SDO; see Maddux et al., 2021 for a review). Please report your model with and without controls. To be clear, I think it is okay if your effects change with controls included – but disclosing that to readers is important.
6. Did you test your model with the two aspects of SDO separated? You mention that the SDO scale has two dimensions, so I was wondering if both dimensions “worked” as IVs, or whether the effects are driven more by SDO-O or SDO-E.
7. I think it would help readers for you to describe the control variables more clearly. For example, what were the specific items for religion, # of trips abroad, and multicultural experience training? I see in Table 1 you report some of the values, but I do not understand what “multicultural experience training” referred to, whether all respondents who said “yes” to the religion question were the same religion, and whether “trips abroad” disentangled living vs. working vs. studying vs. travelling abroad.
8. Could you please include a sample item for each measure used? That would help readers better understand what precisely is being measured, particularly if they are not familiar with the scale cited.
9. To my theoretical points, did you test models with different orders of the variables in your model? For example, whether multicultural acceptance mediated the effect between SDO and stereotyping? I think these results could help address my theoretical point, and be interesting to readers, regardless of the outcome.
MINOR POINTS:
10. Was the sample size 402 or 215? It reads to me like you recruited 402 participants, but the final sample is 215 (who passed all attention checks etc.), so I’d report this as 215 (not 402) in the abstract etc.
11. I found the phrasing of the title a bit confusing. Shouldn’t it be phrased as “Exploring how stereotype modification mediates **THE RELATIONSHIP BETWEEN** social dominance **ORIENTATION** and multicultural acceptance”?
12. This might be stylistic preference, but I found the abstract to bury your key points. I think it would help readers more immediately see the compelling points of your paper if you re-structured it so that the methods points appear at the end (vs. at the beginning). For example, I found the sentences “We used SPSS 26.0 …. with 95% confidence interval (CI)” to not be necessary in the abstract, or at least not be as important as the sentences that followed (which describe the purpose/key findings).
13. On p.3, you write “Research in diverse xxx” – is this a typo? I was confused what “xxx” referred to here.
14. On p. 6, you refer readers to a Table, but do not specify the Table number. I think this is meant to suggest Table 4?
15. I hope you find my comments helpful. I really enjoyed reading your paper and found the content interesting. I tried to be as thorough as possible in the spirit of helping you with your work. Despite my concerns, I do wish you the best of luck with your work.
References not included in the manuscript:
Affinito, S. J., Antoine, G. E., Gray, K., & Maddux, W. W. (2023). Negative multicultural experiences can increase intergroup bias. Journal of Experimental Social Psychology, 109, 104498.
Hodson, G., Crisp, R. J., Meleady, R., & Earle, M. (2018). Intergroup contact as an agent of cognitive liberalization. Perspectives on Psychological Science, 13(5), 523-548.
Maddux, W. W., Lu, J. G., Affinito, S. J., & Galinsky, A. D. (2021). Multicultural experiences: A systematic review and new theoretical framework. Academy of Management Annals, 15(2), 345-376.
Tadmor, C. T., Hong, Y. Y., Chao, M. M., Wiruchnipawan, F., & Wang, W. (2012). Multicultural experiences reduce intergroup bias through epistemic unfreezing. Journal of personality and social psychology, 103(5), 750.
Comments on the Quality of English Language
I thought the English language was totally fine, but there were a few typos that I pointed out in my review that I think are important to address for clarity.
Author Response

(The authors gave the same response as above.)
